# Report-Concept Textual-Prompt Learning for Enhancing X-ray Diagnosis

## ABSTRACT

Despite significant advances in image-text medical visual language modeling, the high cost of fine-grained annotation of images to align radiology reports has led current approaches to focus primarily on semantic alignment between the image and the full report, neglecting the critical diagnostic information contained in the text. This is insufficient in medical scenarios demanding high explainability. To address this problem, in this paper, we introduce radiology reports as images in prompt learning. Specifically, we extract key clinical concepts, lesion locations, and positive labels from easily accessible radiology reports and combine them with an external medical knowledge base to form fine-grained self-supervised signals. Moreover, we propose a novel Report-Concept Textual-Prompt Learning (RC–TPL), which aligns radiology reports at multiple levels. In the inference phase, report-level and concept-level prompts provide rich global and local semantic understanding for X-ray images. Extensive experiments on X-ray image datasets demonstrate the superior performance of our approach with respect to various baselines, especially in the presence of scarce imaging data. Our study not only significantly improves the accuracy of data-constrained medical X-ray diagnosis, but also demonstrates how the integration of domain-specific conceptual knowledge can enhance the explainability of medical image analysis. The implementation code will be publicly available. The implementation code will be available at https://anonymous.4open.science/r/RC-TPL

## CCS CONCEPTS

• **Applied computing → Health informatics**.

## KEYWORDS

X-ray Diagnosis, Vision-Language models, Prompt Learning, Multi-modality

## 1 INTRODUCTION

Deep learning models have demonstrated remarkable potential in the analysis of chest X-rays (CXR), such as disease classification [9]. However, the training of these models relies on a large number of annotated images, which limits their application in practical settings [21]. Pre-trained language models, such as BERT [17], have provided new perspectives for medical vision and language pre-training [24, 37]. Recently, large-scale visual-language pre-training

*ACM MM, 2024, Melbourne, Australia*
© 2024 Copyright held by the owner/author(s). Publication rights licensed to ACM.
ACM ISBN 978-x-xxxx-xxxx-x/YY/MM
https://doi.org/10.1145/nnnnnnn.nnnnnnn

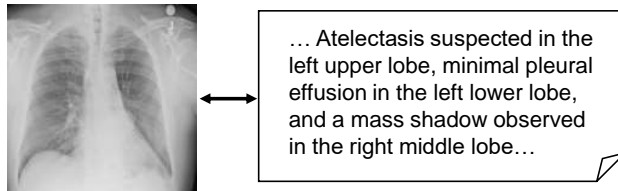

(a) Unexplainable report-level semantic alignment

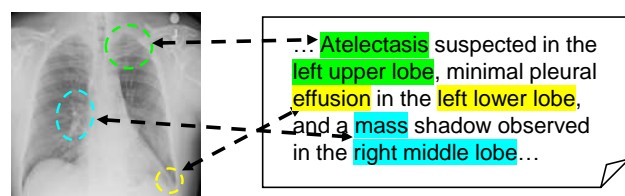

(b) Explainable concept-level semantic alignment

**Figure 1: A comparison between a coarse-grained image-report pairing (a) and a fine-grained image-concept pairing (b). Radiologic reports contain a rich set of information that reflect the specific pathologic and anatomic status of the radiographic image. Semantic alignment using these fine-grained clinical concepts enables explainable diagnosis.**

(VLP) models that utilize naturally paired image-text contrastive learning, such as CLIP [24] and BLIP [18], have made significant progress in visual tasks. Nevertheless, transforming foundational vision-language pre-training models into actual clinical applications remains challenging.

First, the limited availability of medical imaging data presents a challenge for language models to comprehend free-form reports [3]. Unlike natural images that can be annotated with relative ease, medical image annotation requires verification by experienced domain experts, such as clinical physicians, resulting in high thresholds and costs. This leads to a scarcity of annotated medical images, while an abundance of unpaired radiology reports remain underutilized. Although MedCLIP [33] learns from these unpaired texts, there is still a necessity for image involvement. The majority of current medical VLP models merely pair CXR images directly with raw radiology reports, as shown in Figure 1 (a), neglecting the modeling of fine-grained relationships between the semantic content in images and the concepts in reports. Despite efforts to denoise radiology reports and integrate multi-source medical knowledge into pre-training [5, 20, 34], overcoming the limitations of coarse-grained knowledge and reliance on image data remains elusive. Thirdly, explainability is critical. Linking diagnostic outcomes with visual evidence is crucial for aiding radiologists' comprehension of the system and building trust between patients and technology.

To address the above challenges, we advocate learning from prompts by treating radiology reports as medical images. This bypasses the need for time-consuming and laborious image annotation. It is feasible as in pre-trained visual-linguistic models [14] where an image encoder and a text encoder encode images and text into a shared space. For a given image and its radiology report, the visual features produced by the image encoder will be close to the text features of the radiology report produced by the text encoder. This has also been demonstrated on generalized visual multi-classification tasks [8]. Second, when target task data are constrained, prompt learning as a representative parametric learning paradigm has emerged and served as an effective way to adapt VLP models to downstream tasks. For example, CoOp [40] learns from annotated images. DualCoOp [27] learns from annotated images by training positive and negative cue pairs with partially labeled images.

Radiology reports usually contain information from two perspectives, anatomy and pathology, as in Figure 1 (b). Pathologic information is the clinical concept of the chest X-ray examination, e.g., Mass. Anatomical structure describes the anatomical structure and location, e.g., right middle lobe. A recent work, MedKLIP [34], provides a promising approach for decomposing textual information into anatomical and pathological dimensions, but their approach does not have explainable analysis. In this paper, we extract key anatomical and pathological information, including clinical concepts, lesion locations, and positive labels, from easily accessible radiology reports and combine them with an external medical knowledge base to form fine-grained self-supervised signals. Furthermore, we propose a novel Report-Concept Textual-Prompt Learning (RC-TPL), which aligns radiology reports at multiple levels. In the inference phase, report-level and concept-level prompts provide rich global and local semantic understanding of X-ray images. Note that although these prompts are learned only from textual descriptions, they can be easily used to classify entire images as well as image blocks during testing (see Figure 1 (b)).

To sum up, the contributions of this work include.

- We propose to use radiology reports as medical images in prompt learning to adapt medical VLP models to X-ray image diagnosis. Radiology reports are easily accessible and their class labels can be directly exported compared to images. Compared to prompting from images, our method is less affected by the problems of image data limitations and labeling restrictions.
- We extract rich pathological and anatomical information from radiology reports and inject medical knowledge enhancing medical VLP from a knowledge-base or LLMs. It can be flexibly applied to medical classification tasks with or without image annotation.
- We present Report-Concept Textual-Prompt Learning (i.e. RC-TPL) to extract both coarse-grained and fine-grained prompt embeddings for enhancing X-ray Diagnosis.
- The experimental results show that our RC-TPL method outperforms state-of-the-art methods by a large margin on several benchmarks (e.g., ChestX-ray14, CheXpert, and COVID-19) without using any labeled images. Meanwhile, it has well explainable ability.

## 2 RELATED WORK

### 2.1 Medical Vision-Language models

Large-scale pre-trained visual language models (VLMs) have been widely applied across various general domains [18, 19]. For instance, the CLIP [24] was trained on a dataset of 400 million image-text pairs collected from the internet, resulting in an effective alignment of visual and textual encoders. However, due to the complexity of medical reports and the scarcity of large-scale medical image-text datasets, pre-trained visual language models in the medical domain are still being explored. To utilize the medical image-text data, the contrastive learning framework has also been extensively adopted [11, 32]. Many studies utilize precisely matched medical image and text data for contrastive learning, such as ConVIRT [36]. In contrast, MedCLIP [33] introduced a novel approach by decoupling the strict pairing relationship between images and texts, which not only expanded the range of available training data but also reduced the risk of false negatives.

Additionally, some strategies incorporate medical knowledge as part of the model input or as guidance during the model training process [4], such as MedKLIP [34], KAD [35], Med-UniC [30], DeV-iDe [20]. Although these methods have facilitated the integration of knowledge in the training of VLMs, there is still a lack of effective means to integrate the fine-grained discriminative knowledge required in the diagnostic process.

### 2.2 Prompt learning

Prompt learning developed from the field of natural language processing which is motivated by the use of pre-trained language models as a knowledge base from which valuable information can be elicited for downstream tasks using prompt templates [38]. Recently, prompt learning has become a parameter-efficient way to transfer pre-trained knowledge to downstream tasks in data-limited environments [6, 15]. CoOp [40] learns prompts using a few annotated images per category in the target dataset. CoCoOp [39] further improves upon CoOp by formulating prompts in an image-conditional manner. CoCoOp [39] achieves impressive zero-shot inference performance for previously unseen categories under low-resource conditions. In the medical domain, Qin et al. [23] developed a method for automatically generating medical prompts to enhance the knowledge transfer capability of pre-trained visual language models for medical object detection. MedPrompt [38] enables low-cost medical image classification through prompt learning. Most related to our work is XCoOp [2], which leverages medical knowledge by aligning the semantics of images and learning fine-grained knowledge through trainable prompts.

Despite the significant performance achieved by existing prompt learning methods on downstream tasks, they typically require images and some class labels as inputs to learn prompts, making them less applicable in the medical context where image annotation is costly. Other research has developed methods that avoid the need for images in the prompt learning process [8, 41]. Guo et al. proposed TaI [8], which treats text as image learning prompts. Inspired by TaI, we utilize readily available radiology report descriptions to learn prompts. After training, the prompts learned in text can be easily applied to test images.

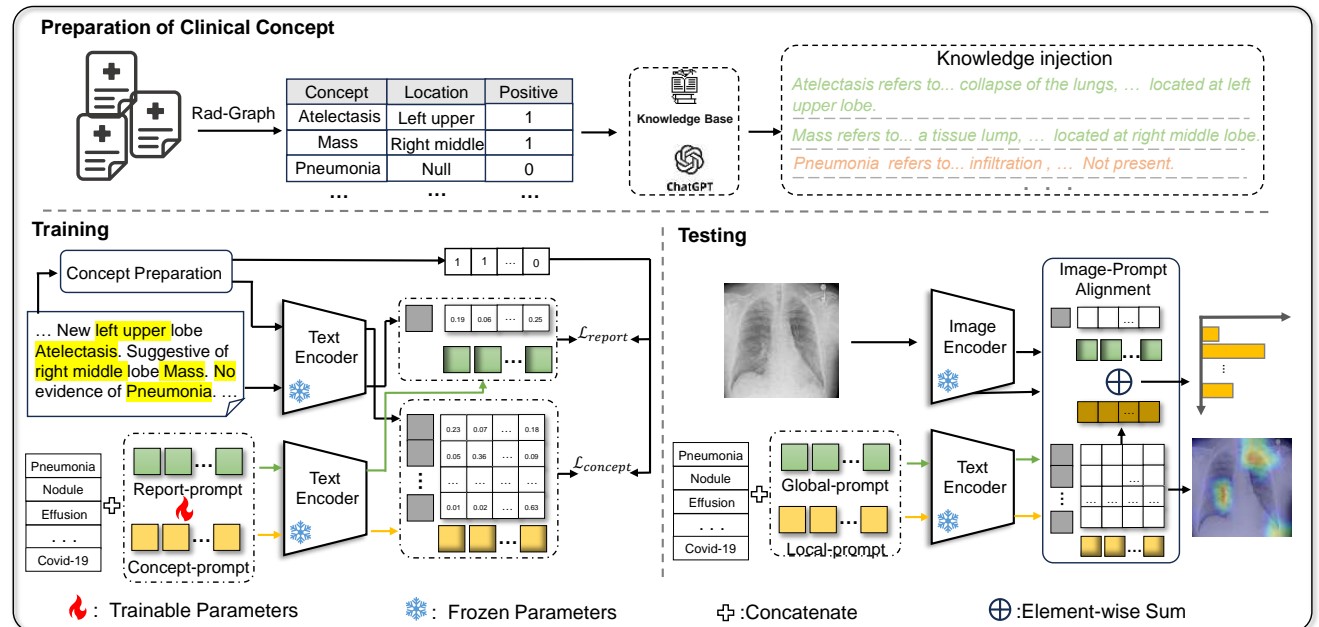

**Figure 2: The pipeline of our approach. The key insight of RC–TPL is using textual descriptions instead of labeled images to train the prompts to enhance the informativeness and explainability of the prompts, guided by multi-granularity concept-based medical knowledge. During training, we use two identical text encoders from the pre-trained MedCLIP to extract report and concept class embeddings, and report text and concept description embeddings from prompts and radiology reports, respectively. During testing, we replace the input from text descriptions to images. The report and concept class embeddings distinguish target classes from global and local image features. The final classification result is obtained by combining the scores of the two branches.**

## 3 METHODOLOGY

In this section, we detail our proposed RC–TPL, including concept extraction and knowledge injection, definition of report-level and concept-level prompts, training and inference. As shown in Figure 2.

### 3.1 Preparation of Clinical Concept

*3.1.1 Concept Extractor.* To obtain sufficient clinical concept information for the identification of X-ray images, we must ensure that the collected reports contain detailed descriptions of diseases. The content of the reports needs to cover all clinical concepts as comprehensively as possible to represent all categories of diseases. To guarantee reproducibility, we use existing radiology reports from public X-ray image datasets (e.g., MIMIC-CXR [16]). It is important to note that although each radiology report is paired with the corresponding image and human-annotated labels, we only utilize the radiology reports themselves and do not use information from images and labels during the training process.

For a target X-ray dataset $\mathcal{D}$ that has a clinical concept set $C = \{c_1, c_2, c_3, \ldots, c_L\}$, Where $L$ denotes the number of concepts and $c_i$ denotes particular concept name like "effusion", "mass", etc. . Given a radiology report $\mathcal{T}$ with multiple sentences, we follow [34] employ the named entity recognition method Rad-Graph[13] to extract (Concept, Location, Positive) triplets from the radiology reports, as shown in the top of figure 2. Here, "Concept"

refers to clinical pathological observation emerge from CXR image (e.g., "Mass"). "Location" indicates the specific anatomical body part related to the clinical observation, such as "left lower lobe." Additionally, there is a "Positive" label to indicate whether the clinical observation reported is positive (present), negative (not present), or uncertain result, which will be used as a supervisory signal for subsequent concept-level semantic alignment. Based on this, we can obtain a set of triplets to describe the fine-grained information from the radiology reports:

$$\mathcal{T}_{\text{concepts}} = (c_i, l_i, \text{pos}_i), i \in \mathcal{N} \tag{1}$$

where $\mathcal{N}$ represents the total number of concepts contained in one radiology report. We only search for radiology reports that contain at least one concept name in $C$.

*3.1.2 Clinical Knowledge Injector.* To achieve fine-grained semantic alignment, we enhance the comprehensibility of the extracted triplets by connecting them with external medical knowledge. Specifically, we query publicly accessible medical knowledge bases (such as Wikipedia [1]) or utilize large language models (such as ChatGPT [2]) to translate clinical concept entities into detailed descriptions. For instance, for "Concept(["MASS"])", we could generate a description like: *"Mass refers to... a tissue lump whose density is significantly different from the surrounding normal tissue..."*. Although this process

---
[1] Wikipedia https://en.wikipedia.org/wiki/
[2] ChatGPT https://chat.openai.com/

may seem simple, transforming entities into descriptions is crucial for more reliable and zero-shot diagnosis, as it breaks down the professional clinical concept into basic attributes shared by various diseases, encouraging the model to deeply understand visual evidence. Regarding the "Position" information, we use prompts such as "It is located in position" to construct sentences. Then, we concatenate these to form a new fine-grained text that guides the learning of concept-level Prompts.

## 3.2 Report-Concept Textual-Prompt Learning

After extensive large-scale image-text contrastive pre-training, textual features have achieved complete consistency with image features of the same semantics [8]. Therefore, radiology text features used to describe specific diagnostic categories also exhibit classifiable recognizability. Based on this alignment of visual-language (VL) representations, we propose a novel perspective that assume the necessity of images for prompt learning, we suggest using textual features describing specific diagnostic categories as substitutes for image features.

Furthermore, medical diagnoses often rely on various clinical concepts (symptoms) observable within specific local regions of images. Considering that different clinical concepts may correspond to different subregions of medical images, we propose Report-Concept Textual-Prompt Learning (i.e., RC-TPL), which uses two sets of prompts to process report-level (i.e., coarsest-grained level) and concept-level (i.e., finer-grained level) features in two parallel branches, respectively. Specific details are shown in Figure 2.

*3.2.1 Report-level Prompt.* Following [8], a report-level is defined as:

$$t_i^R = [\boldsymbol{p}_1, \boldsymbol{p}_2, \boldsymbol{p}_3, \ldots, \boldsymbol{p}_M, c_i] \quad (2)$$

Where $i$ is the concept index, $c_i$ denotes word embedding of the $i$-th concept name. For $j \in \{1, 2, \ldots, M\}$, $\boldsymbol{p}_j$ is a learnable word embedding whose dimension is the same as the dimension of normal word embeddings in the vocabulary. Then the $t_i^R$ is fed to a copy of the text encoder $\mathrm{Enc}_T$ of MedCLIP to generate report-level prompt embeddings for each concept.

$$E^R = \{E_i^R\}_{i=1}^L, E_i^R = \mathrm{Enc}_T(t_i^R) \quad (3)$$

*3.2.2 Concept-level Prompt.* In general, global features are sufficient for common image classification because the target object is usually prominent in the picture. However, in X-ray images, there exists information at the anatomical and pathological levels. Therefore, it motivates us to explore fine-grained features. We propose concept-level prompt definitions as follows:

$$t_i^C = [\boldsymbol{p'}_1, \boldsymbol{p'}_2, \boldsymbol{p'}_3, \ldots, \boldsymbol{p'}_M, c_i] \quad (4)$$

Where $\boldsymbol{p'}_j$ is a learnable word embedding that concatenated with word embedding $c_i$ of the $i$-th concept to obtain the concept prompt. Similarity, the $t_i^C$ is fed to a copy of the text encoder $\mathrm{Enc}_T$ of MedCLIP to generate concept-level prompt embeddings for each concept.

$$E^C = \{E_i^C\}_{i=1}^L, E_i^C = \mathrm{Enc}_T(t_i^C) \quad (5)$$

*3.2.3 Report and concept description embedding.* For radiology reports, we similarly utilize the text encoder in MedCLIP to encode and then take the embedding of the $[CLS]$ token. For each concept,

a series of sentence descriptions obtained after external medical knowledge injection, we encode them separately. Report text embedding and concepts description embedding define as:

$$\boldsymbol{h} = \mathrm{Enc}_T(\boldsymbol{r})^{[CLS]}$$
$$H = \{h_i'^{[CLS]}\}_{i=1}^N, h' = \mathrm{Enc}_T(\mathrm{Knowledge}(c_i)) \quad (6)$$

where $\boldsymbol{r}$ denotes a complete radiology report. $\boldsymbol{h} \in \mathbb{R}^D$ are the extracted global text features. $H \in \mathbb{R}^{L \times D}$ are the extracted concept description features.

Just like in previous methods[40], the prompts are learned by maximizing the probability of classifying each concept description into its ground-truth class. Then, the report-level and concept-level similarities are computed by:

$$\boldsymbol{p}_i = \langle \boldsymbol{u}, E_i^R \rangle, P_{i,j} = \langle U_j, E_i^C \rangle \quad (7)$$

where $\boldsymbol{u}$ denotes either language feature $\boldsymbol{h}$ in training or visual feature $\boldsymbol{f}$ in testing of image $\boldsymbol{x}$, and $U$ denotes $H$ or the flattened dense image features $F$ coordinately. Information in local branch $P$ is aggregated in a spatially weighted manner:

$$\boldsymbol{p}_i' = \sum_{j=1}^L \frac{\exp(P_{ij}/\tau_c)}{\sum_{j=1}^L \exp(P_{ij}/\tau_c)} \cdot P_{ij} \quad (8)$$

where $\tau_c$ accommodates the extent of focusing on a specific location.

## 3.3 Training

Following [8], We adopt the ranking loss[7] to measure the discrepancy between classification scores and ground-truth labels extracted from report for each concept. Specifically, $\mathcal{L}_{report}$ and $\mathcal{L}_{local}$ are formulated as follows:

$$\mathcal{L}_{report} = \sum_{i \in \{c^+\}} \sum_{j \in \{c^-\}} max(0, m - \boldsymbol{p}_i + \boldsymbol{p}_j),$$
$$\mathcal{L}_{local} = \sum_{i \in \{c^+\}} \sum_{j \in \{c^-\}} max(0, m - \boldsymbol{p}_i' + \boldsymbol{p}_j') \quad (9)$$

where $\boldsymbol{p}$ and $\boldsymbol{p}'$ are report-levle and aggregated concept-level similarities described in above. $m$ is the margin controlling how much higher the similarity score with the positive classes is than with the negative classes. During training, we minimize the overall objective $\mathcal{L} = \mathcal{L}_{report} + \mathcal{L}_{local}$ with frozen text encoders, by optimizing the global and local prompts.

## 3.4 Inference

At inference time, given a test image, we can directly infer the existence of certain concepts/disease, and ground their visual evidence. Given an X-ray image scan $\mathcal{X} \in \mathbb{R}^{H \times W \times 3}$, we can compute the visual features $\boldsymbol{f}$ and flattened dense image features $F$ with a visual backbone

$$\{f, F\} = \mathrm{Enc}_I(\mathcal{X}) \quad (10)$$

where $\boldsymbol{f} \in \mathbb{R}^D$ and $F \in \mathbb{R}^{L_I \times D}$, and $L_I = H \times W$ denotes the flattened spatial dimension of visual feature. Then, the report $\boldsymbol{p}$ and concept $\boldsymbol{p}'$ similarities are computed and combined to obtain the final classification score.

# 4 EXPERIMENT

## 4.1 Experimental Setup

### 4.1.1 Pre-training Dataset.

**MIMIC-CXR**. [16] is a large chest X-ray database composed of more than 227k paired image-report data research, collected from 65,379 different scans of patients in the Beth Israel Deaconess Medical Center in Boston. It is important to note that. We only use the freely available text radiology reports of this dataset for prompt learning in training. For better training, we excluded lateral views and reports that included fewer than two concepts, resulting in a pre-training dataset with 223,673 MIMIC-CXR image-text pairs.

### 4.1.2 Datasets for Downstream Tasks.

**CheXpert**. [12] is a dataset containing 224,316 chest X-rays with 14 observation labels, covering 65,240 patients who underwent radiological examinations at Stanford Medical Center.

**ChestX-ray14**. [31] consists of 112,120 frontal x-ray images with 14 common diseases labeled from 30,805 patients collected by the National Institutes of Health (NIH) in the United States.

**COVIDx CXR-3 and COVID Rural**. [22, 28] aim to evaluate on diagnosing COVID-19. COVIDx CXR-2 contains 29,986 X-rays with the COVID-19 classification labels. We used it as a classification dataset. In addition, we use the COVID Rural dataset for COVID-19 segmentation, which contains more than 200 chest radiographs with segmentation masks.

**RSNA Pneumonia**. [26] is a chest X-ray image dataset publicly provided by the NIH, which consists more than 260k chest X-rays with corresponding pneumonia opacity masks. It can be used as both a classification and grounding task.

**SIIM Pneumothorax**. is a more than 12k frontal chest radiograph dataset with pneumothorax mask collected by the Society for Imaging Informatics in Medicine and the American College of Radiology. Similar to the RSNA pneumonia dataset, it can be used as both a classification and grounding task.

### 4.1.3 Baselines.
We evaluate our method against a range of state-of-the-art (SOTA) methods for medical VLP, including **ConVIRT** (2022) [36], **GLORIA** (2021) [11], **MedCLIP-ViT** (2022) [33], **BioViL** (2022) [3], **CheXzero** (2022) [29] **MedKLIP** (2023) [34], **MedPrompt-ViT** (2024) [38], **MeDSLIP** (2024) [5].

**ConVIRT** trains two modality-specific encoders by bidirectional contrastive loss to learn visual representations. **GLoRIA** utilizes both global and fine-grained features for medical VLP. **MedCLIP-ViT** decouple images and texts for multimodal contrastive learning, and we choose ViT as the image encoder. **BioViL** proposes a radiology-specific text encoder for the subsequent classical pipeline of VLP. **CheXzero** retrains one CLIP model with a corpus of the medical domain. **MedKLIP** designs one novel entity extraction and transition module to inject domain-specific knowledge into the process of VLP. **MedPrompt-ViT** uses a weakly supervised prompt learning method to automatically generate medical prompts. **MeDSLIP** establishes vision-language fine-grained alignments via disentangling visual and textual representations.

### 4.1.4 Metrics.
**AUC** and **ACC** are measured for classification tasks. **Pointing Game** and **Dice** are used for evaluating the grounding performance. In specific, we extract the region with max response in the output heat-map, for one instance, if the region hit the ground-truth mask, it is considered a positive prediction, otherwise negative. Finally, accuracy can be calculated as the pointing game score. We report the maximal Dice score for each model. **Precision** and **Recall** refer to the detection Precision and Recall. For medical, in some hard cases, especially for the zero-shot setting, Dice may be too strict to reflect the performance difference. Precision and recall scores can compensate for these. **Precision@$K$** is used for measure the performance of various models in Image-Text retrieval task.

### 4.1.5 Implementation.
We adopt MedCLIP [33] with ResNet-50 [10] as the image encoder, and use the MedCLIP with BioClinical-BERT [1] as the text encoder. During training, the parameters of the two encoders are kept frozen, and only learnable prompts are optimized. Our learnable prompts are shared between classes across all datasets. Following [8], we used Gaussian noise sampled from $\mathcal{N}(0.02)$ to initialize the value of each prompt parameter. The length of both the report prompts and the conceptual prompts are set to $M = 12$, while longer sequences bring negligible improvement. We pre-train on the MIMIC-CXR dataset involves a batch size of 64, an AdamW optimizer with learning rates $2e-5$, and a cosine scheduler. And for all datasets, the number of training epochs is set to 20 on 4×V100 GPUs.

## 4.2 Experimental Results

### 4.2.1 Zero-shot Classification.
We conduct zero-shot image classification evaluation on four datasets: ChestX-ray14, CheXpert, COVID, and RSNA. Through matching the encoded image embedding and the embeddings of learned report-level and concept-level prompts for each disease class to achieve zero-shot prediction. We illustrate the results in Table 1. We analyze and discuss separately the categories that were seen in the pre-training and the categories of chest diseases that were not seen.

**Seen Diseases.** The pre-training dataset MIMIC-CXR includes 14 disease labels. The specific categories are consistent with the disease categories included in CheXpert, as listed in Figure 3 (right). On both the CheXpert and RSNA pneumonia datasets, their diseases were seen in the pre-training text. Despite the fact that the images were collected by different hospitals at the time, our model improves the AUC from 0.83 to 0.85 on CheXpert, and from 0.86 to 0.89 on the RSNA dataset. For the ChestX-ray14 dataset, which has half of the disease labels not included in MIMIC-CXR, our model improves the AUC from 0.78 to 0.83. This suggests that our approach can better handle multi-center, multi-disease data distribution in medical.

**Unseen Diseases.** We are considering a strict set for open-set classification, in particular, we utilize Covid-19 to evaluate the model. Covid is a new disease that only appeared in 2019, and the MIMIC-CXR reports collected in 2015 do not include any Covid entities, so it requires the model to have the ability to diagnose truly unseen diseases. As shown in Table 1, it is difficult to make a correct diagnosis with existing methods that rely only on the name of the disease. In contrast, methods that introduce medical

**Table 1: Comparison with other state-of-the-art methods on zero-shot classification task. AUC and ACC scores are reported. For ChestX-ray14 and CheXpert, the metrics all refer to the macro average on the all diseases. ∗ represents partial overlap of disease categories seen in the pre-training data, and ∗∗ represents disease categories that did not seen in the pre-training data.**

| Methods | ChestX-ray14* | | CheXpert | | COVID** | | RSNA | |
|---|---|---|---|---|---|---|---|---|
| | AUC↑ | ACC↑ | AUC↑ | ACC↑ | AUC↑ | ACC↑ | AUC↑ | ACC↑ |
| ConVIRT | 0.6101 | 0.7102 | 0.6640 | 0.7229 | 0.6208 | 0.6093 | 0.8042 | 0.7611 |
| GLoRIA | 0.6608 | 0.7616 | 0.6872 | 0.7472 | 0.6539 | 0.6090 | 0.7145 | 0.7129 |
| MedCLIP-ViT | 0.6696 | 0.7008 | 0.7222 | 0.7698 | 0.6221 | 0.5361 | 0.7260 | 0.7311 |
| BioViL | 0.6910 | 0.7844 | 0.5768 | 0.6446 | 0.5538 | 0.5375 | 0.8280 | 0.7669 |
| CheXzero | 0.7263 | 0.8278 | 0.7529 | 0.8119 | 0.6556 | 0.6578 | 0.8577 | 0.7942 |
| MedKLIP | 0.7628 | 0.8645 | 0.8389 | 0.8680 | 0.7396 | 0.7009 | 0.8693 | 0.7998 |
| MedPrompt-ViT | 0.7604 | 0.8532 | 0.8256 | 0.8087 | 0.6853 | 0.6701 | 0.8386 | 0.7884 |
| MeDSLIP | 0.7834 | 0.8883 | 0.8339 | 0.8572 | 0.7556 | 0.7267 | 0.8649 | **0.8098** |
| Ours | **0.8327** | **0.8906** | **0.8518** | **0.8732** | **0.7732** | **0.7651** | **0.8888** | 0.8072 |

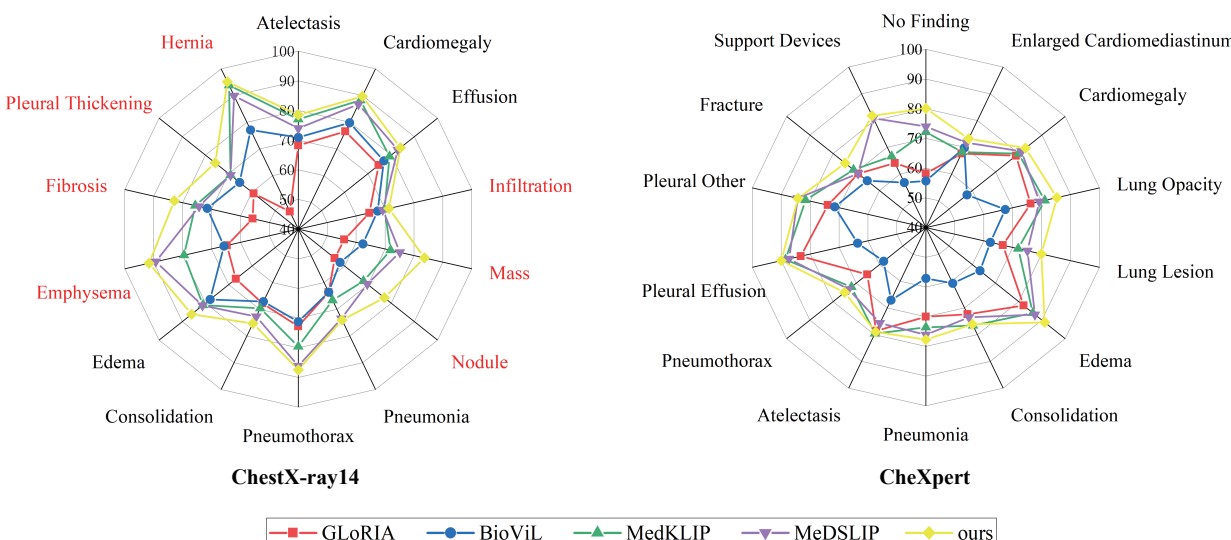

**Figure 3: Fine-category performance of different methods on ChestX-ray14 (left) and CheXpert (right). AUC scores of each category are displayed. Red font indicates that the category is not in the MIMIC-CXR pre-training data and is considered an unseen disease.**

knowledge including MedKLIP, MeDSLIP and our method can substantially improve the model performance. Our method improves the introduction of fine-grained concept descriptions that can be generalized to new chest diseases, since images of Covid-19 also show similar pathological features such as "Lung opacity". Compared to state-of-the-art methods that do not incorporate medical knowledge, our method significantly improves the performance of AUC from 0.68 to 0.77 and ACC from 0.67 to 0.76, suggesting that the introduction of fine-grained concept medical knowledge is crucial for unseen diseases classification. This is also confirmed on unseen diseases in the ChestX-ray14 dataset, as shown in the disease classification performance highlighted in red font in Figure 3 (left).

*4.2.2 Zero-shot Region Grounding.* Beyond plain diagnosis, explainability is equally crucial in healthcare to improve the reliability and trustworthiness of machine learning systems. Here, we consider providing explainability by grounding anomalies in predictions and compare it with existing methods to demonstrate the effectiveness of our proposed concept prompt learning for explainability. Similarly, we split the diseases into seen and unseen ones, depending on whether their names have peared in the medical reports. Specifically, "Pneumonia" and "Pneumothorax" are treated as seen, and "Covid-19" is treated as unseen.

***Seen Diseases.*** Tabel 2 and Table 3 shows the results on RSNA Pneumonia dataset and SIIM-ACR Pneumothorax dataset respectively. Follow [34], we only consider pointing game, recall, and precision, as the pneumothorax region tends to be thin and narrow, and grounding is more challenging. Our approach leads to

better performance on different metrics, especially on fixed-point game scores, especially on the pointing game score. Other baseline methods can not realize this function. As shown in Table 2, our proposed model outperforms existing methods in all metrics, such as increasing the fixed-point game score from 0.88 to 0.90. Overall, our method can achieve better performance than previous methods.

**Unseen Diseases.** As shown in Table 4, we also conducted a zero-shot grounding experiment on the unseen disease Covid-19. Our model shows consistent improvements across all metrics, e.g., increasing the pointing game score from 0.54 to 0.62. Our model with knowledge-enhanced concept-level prompt embedding facilitates the iamge encoder to learn potential evidence related to the disease, and therefore produces more explainable representations compared to previous methods.

**Table 2: Comparison with other state-of-the-art methods for the zero-shot region grounding tasks on RSNA pneumonia dataset.**

| Methods | Pointing Game↑ | Recall↑ | Precision↑ | Dice↑ |
|---------|----------------|---------|------------|-------|
| GLoRIA | 0.7607 | 0.8330 | 0.1621 | 0.3468 |
| BioViL | 0.8342 | 0.8521 | 0.5034 | 0.4386 |
| MedKLIP | 0.8721 | 0.8661 | 0.6320 | 0.4649 |
| MeDSLIP | 0.8857 | 0.8682 | 0.6471 | 0.4955 |
| Ours | **0.9011** | **0.8732** | **0.6920** | **0.5087** |

**Table 3: Comparison with other state-of-the-art methods for the zero-shot region grounding tasks on the SIIM-ACR Pneumothorax dataset.**

| Methods | Pointing Game↑ | Recall↑ | Precision↑ |
|---------|----------------|---------|------------|
| GLoRIA | 0.0651 | 0.2377 | 0.0585 |
| BioViL | 0.0252 | 0.1963 | 0.1429 |
| MedKLIP | 0.1975 | 0.3562 | 0.1940 |
| MeDSLIP | 0.2278 | 0.3632 | 0.1962 |
| Ours | **0.2531** | **0.3825** | **0.2071** |

**Table 4: Comparison with other state-of-the-art methods for the zero-shot region grounding tasks on covid-19 opacity region grounding task.**

| Methods | Pointing Game↑ | Recall↑ | Precision↑ |
|---------|----------------|---------|------------|
| GLoRIA | 0.2727 | 0.2821 | 0.1336 |
| BioViL | 0.1818 | 0.2393 | 0.1637 |
| MedKLIP | 0.5818 | 0.5214 | 0.4959 |
| MeDSLIP | 0.5436 | 0.5362 | 0.4421 |
| Ours | **0.6214** | **0.5833** | **0.5018** |

*4.2.3 Zero-shot Image-Text Retrieval.* To better validate the effectiveness of our proposed method in text senmantic alignment. We followed the experimental setup in MedCLIP [33] and evaluated the semantic richness of the representations learned by the model through image-text retrieval tasks in CheXpert testing split of five disease categories, including Atelectasis, Cardioomegaly, Edema, Pleural, and Effusion. Due to the lack of publicly available report data for CheXpert, we used the MIMIC-CXR dataset to generate reports/sentences. We sampled 200 sentences for each of the five classes present in the CheXpert-5x200 dataset. This generated 1000 images and 1000 sentences as the retrieval dataset. The results are shown in Table 5. Our method achieved better performance, which proves that it effectively provides the semantic information required for retrieving text.

**Table 5: Results of Image-Text retrieval tasks on CheXpert5x200 dataset.**

| Methods | P@1↑ | P@2↑ | P@5↑ | P@10↑ |
|---------|------|------|------|-------|
| ConVIRT | 0.20 | 0.20 | 0.20 | 0.21 |
| GLoRIA | 0.47 | 0.47 | 0.46 | 0.46 |
| MedCLIP-ViT | 0.46 | 0.49 | 0.49 | 0.51 |
| MedKLIP | 0.49 | 0.52 | 0.50 | 0.55 |
| Ours | **0.52** | **0.58** | **0.59** | **0.61** |

## 4.3 Ablation studies

*4.3.1 The number of prompts $M$.* We conduct ablation studies on CheXpert following the same setting as in Table 1. The result in Figure 4 (left) shows that the model achieves the best performance when $M$ = 12. When the prompt length is too long, both the training efficiency and the computation budget will be increased.

Further, we performed ablation experiments on Report-level and Concept-level at $M$=12. From Figure 4 (left), we can conclude that our recept-level can achieve better results by injecting external medical knowledge.

*4.3.2 Knowledge injection compare.* We further explored different knowledge injection approaches in the preparation of the concept description phase. Specifically, we used five concept knowledge injection settings, including without knowledge (W/O KG), general knowledge (i.e., a fixed format for knowledge without a specific clinical concept, e.g., a photo of [concept name].) , medical knowledge generated based on Large Language Models (LLMs) (LLMs KG), medical knowledge without anatomical location based on clinical knowledge base (KB KG W/O position), and clinical knowledge base knowledge with anatomical location (KB KG). Similarly, the previous three knowledge injection settings splice anatomical location information. Specifically, we take the unseen Covid-19 dataset as an example. Figures 4(middle) and 4 (right) show that without external medical knowledge or general knowledge leads to degraded model performance. Secondly, without incorporating the disease concept in the anatomical location information of the image brings a degradation to the model performance , especially in the grounding task. Finally, we observed that clinical knowledge generated with LLMs is also comparable to specialized knowledge

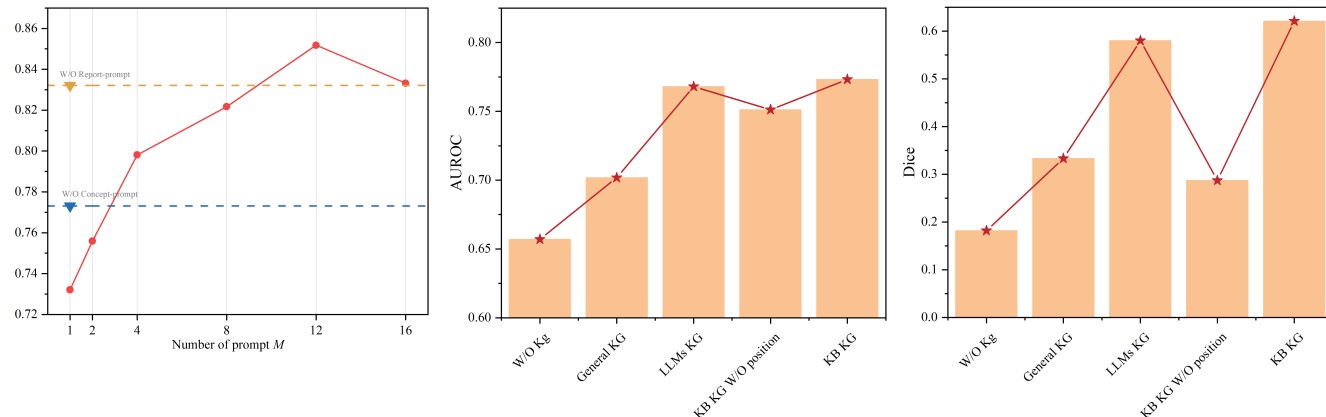

Figure 4: Left image shows the change in AUC of the number of prompts $M$ on the CheXpert dataset, as well as the ablation results for report-prompt and accept-prompt. Middle and right images show the change in performance of classification and grouding tasks for our prompt learning on the unseen Covid-19 dataset for different external knowledge injection methods.

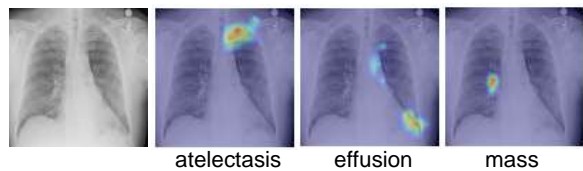

Figure 5: Visualization of the selected disease class with the learned concept prompts in the same image using Grad-CAM.

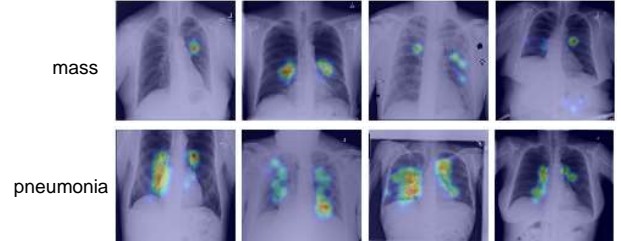

Figure 6: Visualization of the same disease class with the learned concept prompts on different X-ray images using Grad-CAM.

bases. The comprehension capabilities of LLMs can be utilized for more research extensions in the future.

*4.3.3 Visualization of prompts.* To verify that different prompts do reflect different image attributes, we visualize the image contents corresponding to different prompts using Grad-CAM [25].

Specifically, given a test image, a feature map $F$ is obtained by image encoder. Concept-levle prompt and each disease category $i$ are concatenated and then passed through the text encoder to obtain concept prompt embedding $E_i^R$. $F$ and $E_i^R$ then used to calculate similarity $P$, which is adopted to highlight the corresponding image contents using Grad-CAM.

In Figure 5, the image contents in different columns correspond to the activation regions of different disease categories. As can be seen in Figure 5, for the same X-ray image, concept prompts with different disease category do reflect different regions in the image, indicating the validity and diversity of the learned prompts.

To verify whether the learned concept prompts indeed reflect image attributes with high-level semantics, we visualize the content of the same disease category (e.g., mass) on different images in in Figure 6. It can be seen that our method can focus on different anatomical locations of the same disease category on different images. This suggests that the concept prompts effectively learns key attributes that can be generalized across images, thereby improving the performance of X-ray diagnostics.

## 5 CONCLUSION

In this paper, we propose a new perspective on prompt learning that treats radiology reports as X-ray images, which learns prompts from the discriminative features of concept descriptions.. Compared to previous prompt learning methods trained with images, our method benefits from the easy accessibility of scalable content-rich radiology reports, which enables prompt learning for visual diagnostic tasks even without downstream X-ray data. Report-level and concept-level prompts are introduced to leverage global and fine-grained features for better X-ray diagnosis ability. Our approach outperforms other prompt learning methods while retaining the inherent explainability of visual and textual interpretation. Extensive experiments and explainability analyses on variety of datasets show that our method achieves both excellent performance and explainability.

## ACKNOWLEDGMENTS

Thank all reviewers, PC members, and ACs for their time and efforts.

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
