# OpenReview forum: "Report-Concept Textual-Prompt Learning for Enhancing X-ray Diagnosis"
_acmmm.org/ACMMM/2024/Conference — MM2024 Poster_

### Official Review · Reviewer_F6v1 · 2024-05-16

**Rating:** 4
**Confidence:** 3

**Summary:**

This paper proposes a novel Report-Concept Textual-Prompt Learning (RC-TPL) framework for enhancing X-ray diagnosis by leveraging radiology reports instead of labeled images. The key idea is to extract clinical concepts, lesion locations, and positive labels from radiology reports and combine them with an external medical knowledge base to form fine-grained self-supervised signals. The RC-TPL model aligns radiology reports at both report-level and concept-level, providing rich global and local semantic understanding for X-ray images during inference.

**Strengths:**

1. The proposed method addresses the challenge of limited availability of annotated medical imaging data by utilizing readily accessible radiology reports for prompt learning.
2. The integration of domain-specific conceptual knowledge from radiology reports and external medical knowledge bases enhances the explainability of medical image analysis.
3. The RC-TPL framework aligns radiology reports at multiple granularity levels (report-level and concept-level), enabling both global and local semantic understanding of X-ray images.
4. The experimental results demonstrate superior performance compared to various baselines, especially in data-constrained scenarios.

**Limitations:**

1. The extraction of clinical concepts relies on the Rad-Graph method, which may have limitations in accurately identifying all relevant concepts. How does the quality of concept extraction affect the overall performance of RC-TPL?
2. The integration of external medical knowledge is described briefly. More details could be provided on how the knowledge is queried and incorporated into the concept descriptions. Is there a systematic way to ensure the quality and relevance of the injected knowledge?
3. In Equation 6, the concept description embedding $H$ is defined as $H = {h'{i}^{[CLS]}}^N{i=1}$. However, it is unclear how the dimension of $H$ becomes $\mathbb{R}^{L\times D}$ as mentioned later. Please clarify the dimensionality of $H$.
4. The aggregation of local branch information in Equation 8 uses a softmax-like operation. Is there any ablation study to justify the choice of this aggregation method compared to other alternatives?
5. The paper mentions combining the scores from report-level and concept-level branches for the final classification. However, the specific combination method is not provided. Is it a simple averaging, weighted sum, or some other approach?
6. The experimental setup and datasets used for evaluation could be described in more detail to enhance reproducibility.

**Suitability:**

2

---

### Official Review · Reviewer_beMR · 2024-05-22

**Rating:** 2
**Confidence:** 4

**Summary:**

The authors propose to replace X-ray images with medical reports to conduct prompt learning, which aims to address the problem of data limitation. To achieve this, a knowledge injection method and a Report-Concept Textual-Prompt Learning approach are introduced. Experimental results verify the effectiveness of the proposed model for improving X-ray Diagnosis.

**Strengths:**

This paper is well-written, with clear descriptions that make it easy for readers to understand the ideas, methods, and results. The authors have conducted necessary experiments to demonstrate the effectiveness.

**Limitations:**

1. The paper lacks innovation; the method of extracting fine-grained medical knowledge from reports is quite common. Additionally, the approach of using reports instead of images for training is very similar to [1]. The report-prompt and concept-prompt in this paper are almost identical to the global-prompt and local-prompt in [1], and even the figures are drawn in a similar style.

[1] Guo Z, Dong B, Ji Z, et al. Texts as images in prompt tuning for multi-label image recognition[C]//Proceedings of the IEEE/CVF Conference on Computer Vision and Pattern Recognition. 2023: 2808-2817.

2. One major concern: Unlike CLIP for natural image-text pairs, MedCLIP, despite improving the correlation between medical images and reports, could be limited by the limited scale of the medical data. So, it seems not as effective as CLIP to align the embeddings of the two modalities in the same semantic space (maybe for multi-modal medical data, the learned spaces are close, but still have a huge gap). Therefore, directly replacing X-ray images with reports might not be a reasonable approach. It is recommended to qualitatively or quantitatively verify the similarity between image embeddings and report embeddings to support your idea.

3. The definition of module name in Figure 2 and the Method section is inconsistent. For example, the Concept Extractor in 3.1.1 and the Clinical Knowledge Injector in 3.1.2 do not have corresponding names in the figure. Additionally, there are numerous typos, such as the double “.” in line 908 and the repeated sentences in lines 29-31.

**Suitability:**

3

---

### Official Review · Reviewer_ojB2 · 2024-05-24

**Rating:** 4
**Confidence:** 2

**Summary:**

To avoid relying on high-cost fine-grained annotations while focusing on critical diagnostic information in reports, this paper introduces radiology reports as images into rapid learning. Key clinical concepts, lesion locations, and positive labels are extracted from easily accessible radiology reports and combined with external medical knowledge bases to form fine-grained self-supervised signals. A novel Report-Concept Textual-Prompt Learning (RC-TPL) method is proposed, aligning radiology reports at multiple levels and achieving state-of-the-art performance across various downstream tasks.

**Strengths:**

1. This paper identifies the challenges of high-granularity alignment between radiology images and reports, emphasizing that existing methods have not considered the critical diagnostic information within reports, an issue that has not been previously addressed.

2. The proposed method introduces radiology reports as prompts in learning, demonstrating both innovation and rationale. By extracting key information from reports and integrating it with external knowledge bases to create fine-grained annotations, the approach effectively enhances cross-modal alignment performance.

3. Comprehensive experiments, including comparisons with state-of-the-art methods and ablation studies, provide evidence for the effectiveness of the proposed method.

**Limitations:**

1. The implementation details of the Clinical Knowledge Injector should be provided.
2. Ablation studies with different backbones should be included.
3. Figure 4 should be represented more effectively.

**Suitability:**

3

---

### Official Review · Reviewer_8DTU · 2024-05-25

**Rating:** 5
**Confidence:** 3

**Summary:**

This paper proposes a novel method to use radiology reports as medical images for cue learning, aiming to solve the problem of high cost of fine-grained annotation and improve the interpretability of medical image analysis. By extracting key clinical concepts, lesion locations, and positive labels from radiology reports and combining them with external medical knowledge bases to form fine-grained self-supervised signals, this paper proposes a Report-Concept Text Prompt Learning (RC-TPL) method. Experiments on multiple X-ray image datasets show that RC-TPL performs well under data-scarce conditions, significantly improves diagnostic accuracy, and demonstrates how the integration of domain-specific conceptual knowledge enhances the interpretability of medical image analysis.

**Strengths:**

Innovation: This paper proposes an innovative method of treating radiology reports as images for prompt learning, bypassing the high-cost image annotation problem.

Interpretability: Through multi-granularity prompt learning, the model not only improves diagnostic accuracy, but also enhances the interpretability of diagnostic results, which is particularly important in the medical field.

Detailed experiments: This article conducted extensive experiments on multiple data sets and verified the effectiveness of the method through a variety of evaluation indicators. The results are convincing.

Comparative analysis: A variety of state-of-the-art medical visual language pre-training methods are compared, and the superiority of this method is demonstrated in detail.

**Limitations:**

Insufficient explanation of external knowledge bases: Although this article mentions incorporating external medical knowledge bases, the discussion of knowledge base selection, quality, and coverage is not detailed enough. Furthermore, the insufficient presentation of visualization results from different knowledge bases makes it difficult to assess the specific role of various knowledge bases in improving model performance and interpretability.

Versatility: Although the method has been validated on multiple X-ray data sets, it is not yet clear whether it is applicable to other medical images (such as MRI, CT).

**Suitability:**

2

---

### Meta-Review · Area_Chair_DuvL · 2024-07-02

**Recommendation:** Accept (Poster)
**Confidence:** 4

**Metareview:**

Overall, all reviewers are satisfied with the response given by the authors, and are glad to see that the quality of the paper has been improved substantially.